# Serum CS/DS, IGF-1, and IGFBP-3 as Biomarkers of Cartilage Remodeling in Juvenile Idiopathic Arthritis: Diagnostic and Therapeutic Implications

**DOI:** 10.3390/biom14121526

**Published:** 2024-11-28

**Authors:** Katarzyna Winsz-Szczotka, Kornelia Kuźnik-Trocha, Ewa M. Koźma, Bogusław Żegleń, Anna Gruenpeter, Grzegorz Wisowski, Katarzyna Komosińska-Vassev, Krystyna Olczyk

**Affiliations:** 1Department of Clinical Chemistry and Laboratory Diagnostics, Faculty of Pharmaceutical Sciences in Sosnowiec, Medical University of Silesia, ul. Jedności 8, 41-200 Sosnowiec, Poland; kkuznik@sum.edu.pl (K.K.-T.); mkozma@sum.edu.pl (E.M.K.); zeglenb@wp.pl (B.Ż.); vis@sum.edu.pl (G.W.); kvassev@sum.edu.pl (K.K.-V.); olczyk@sum.edu.pl (K.O.); 2Department of Rheumatology, The John Paul II Pediatric Center in Sosnowiec, ul. Gabrieli Zapolskiej 3, 41-218 Sosnowiec, Poland; anna.gruenpeter@gmail.com

**Keywords:** juvenile idiopathic arthritis, cartilage metabolism, extracellular matrix, chondroitin sulfate, dermatan sulfate, IGF-1/IGFBP-3 axis, diagnostic biomarkers, inflammation

## Abstract

Cartilage destruction in juvenile idiopathic arthritis (JIA) is diagnosed, often too late, on basis of clinical evaluation and radiographic imaging. This case–control study investigated serum chondroitin/dermatan sulfate (CS/DS) as a potential biochemical marker of cartilage metabolism, aiming to improve early diagnosis and precision treatment for JIA. We also measured the levels of insulin-like growth factor-1 (IGF-1) and insulin-like growth factor-binding protein-3 (IGFBP-3) (using ELISA methods) in JIA patients (*n* = 55) both before and after treatment (prednisone, sulfasalazine, methotrexate, administered together), and analyzed their relationships with CS/DS levels. Untreated JIA patients [8.26 µg/mL (6.25–9.66)], especially untreated girls [8.57 µg/mL (8.13–9.78)] and patients with a polyarticular form of the disease [7.09 µg/mL (5.63–8.41)], had significantly reduced levels of serum CS/DS compared with the control [14.48 µg/mL (10.23–15.77)]. Therapy resulted in a significant increase in this parameter, but without normalization. We also found significantly lower levels of IGF-1 [66.04 ng/mL (49.45–96.80)] and IGFBP-3 [3.37 ng/mL (2.65–4.88)] in untreated patients compared with the control [96.92 ng/mL (76.04–128.59), 4.84 ng/mL (4.21–7.750), respectively]. Based on receiver operating characteristic (ROC) curve analysis, the blood concentration of CS/DS demonstrated the highest diagnostic power (AUC = 0.947) for JIA among all the tested markers. Untreated patients showed significant correlations between CS/DS and IGF-1 (r = −0.579, *p* = 0.0000), IGFBP-3 (r = −0.506, *p* = 0.0001), and C-reactive protein (r = 0.601, *p* = 0.0005). The observed changes in CS/DS during the course of JIA, influenced by both impairment of the IGF/IGFBP axis and inflammation, indicate the need for continued therapy to protect patients from potential disability. We suggest that CS/DS may be a useful biomarker of disease activity and could be employed to assess treatment efficacy and progress toward remission.

## 1. Introduction

Juvenile idiopathic arthritis is a complex of biologically diverse immune and inflammatory diseases of the connective tissue, which begin in childhood. This arthropathy develops in children with hereditary disorders of the immune response, the symptoms of which are triggered by external factors such as infections, stress, and unfavorable psychosocial conditions [1,2,3]. As a consequence of these conditions, the breakdown of autoimmunotolerance and the local (joint) hypersecretion of pro-inflammatory cytokines including tumor necrosis factor α (TNF-α), interleukin (IL)-1β, or IL-6 are observed [3,4,5]. Long-term elevated levels of these cytokines lead to chronic inflammation, primarily localized in the joint synovial membranes. Moreover, this process frequently spreads to adjacent tissues and leads to remodeling of all of the musculoskeletal structures, which might partly result from disturbances in the metabolism of the extracellular matrix (ECM) components [3].

The ECM is a multi-component, highly ordered, flexible network structure that fills intercellular spaces. The composition of the ECM typical for a given tissue or organ ensures its integrity and determines unique biophysical properties, such as flexibility, plasticity, viscosity, or tensile strength [6]. The aforementioned biomechanical properties are related to the tissue-specific organization of macromolecules that form the ECM, including structural proteins, i.e., collagens or elastin, as well as non-collagenous glycoproteins, including fibronectin, laminin, entactin, undulin, or thrombospondin. The matrix components include also proteoglycans (PGs), for example aggrecan, decorin, biglycan, versican, perlecan, lumican, or fibromodulin [7]. PGs are glycosylated molecules in which one or more heteropolysaccharide chains of glycosaminoglycans (GAGs) and oligosaccharide chains are attached by O-glycosidic or N-glycosidic bonds to a centrally located polypeptide chain, which constitutes the protein core. The type of the attached GAGs chain determines the structural features and biological properties of the PGs [8]. 

Where GAGs are concerned, galactosaminoglycans (GAAGs) composed of alternating N-acetyl-D-galactosamine residues and hexuronic acid residues, i.e., D-glucuronic or L-iduronic acid, are predominant in tissues. GAAGs include chondroitin–dermatan sulfate (CS/DS). In addition to GAAGs, tissues contain glucosaminoglycans (with N-acetyl or N-sulfate-D-glucosamine and hexuronic acid residues, or neutral hexose, i.e., D-galactose), i.e., heparan sulfates, keratan sulfates, and hyaluronic acid [8]. These ECM components, while remaining in a state of dynamic equilibrium with fibrous proteins and non-structural glycoproteins, both play the role of a static scaffold for cells and constitute a kind of physical barrier that protects multicellular, three-dimensional tissue systems against the introduction of new elements and their excessive loss. Moreover, PGs also function as signaling molecules in the physiological state and under pathological conditions, including inflammation, diabetes, atherosclerosis, and cancer [6,7,8,9,10,11]. In addition, by interacting with the tyrosine kinase receptors and Toll-like receptors, they regulate cell migration, proliferation, apoptosis, and autophagy, as well as innate immunity and angiogenesis [9,10].

As we showed in our previous studies, the metabolism of ECM components undergoes changes in the course of JIA [11,12]. The prevalence of anabolic over catabolic processes, which is characteristic of the growing period, is disturbed in the patients [3]. The increased degradation stimulated by proteolytic and pro-oxidative hyperactivity in children with JIA might be not compensated for by the intensity of biosynthesis of ECM compounds, which involves an insulin-like growth factor 1 (IGF-1). Anabolic growth factor is involved in cartilage development and homeostasis. In addition, IGF-1 also protects the ECM against IL-1 and TNF-α-mediated degradation during joint damage [13]. Cartilage destruction in JIA patients is diagnosed, often too late, on the basis of clinical evaluation and radiographic imaging. At present, there is no easy-to-detect biochemical marker that allows quick diagnosis of this pathological condition. 

In view of the above, and due to the fact that changes in tissue proteoglycans are reflected in the CS/DS profile in the blood, the main goal of this study was to assess serum levels of CS/DS, the most abundant glycan fraction in the blood stream and thus best reflecting changes in tissue PGs in patients with JIA, at the time of clinical manifestation of the disease, as well as in the same patients after treatment that modified the course of the inflammatory process and contributed to clinical improvement. Moreover, it was also decided to evaluate the relationship between the levels of these regulatory molecules and the levels of CS/DS in the blood of children with JIA, due to the significant role of IGF-1 in the regulation of ECM components’ metabolism, which is mediated by IGF-specific binding protein 3 (IGFBP-1) [14]. 

For a more complete assessment of the dynamics of possible changes in CS/DS, IGF-1, and IGFBP-3 levels, it was decided to analyze the influence of sex and the form of the disease on the levels of these compounds in the blood of JIA patients. In order to achieve the main goal of the study, we investigated the relationships between the concentrations of the assessed parameters and the activity of the inflammatory process, expressed as the serum level of C-reactive protein (CRP) and the value of the erythrocyte sedimentation index (ESR).

## 2. Materials and Methods

### 2.1. Subject

Fifty-five Polish Caucasian children of both sexes (forty-one girls, fourteen boys), 7.31 ± 4.86 years of age (mean ± standard deviation) and meeting the International League of Associations for Rheumatology (ILAR) criteria for oligoarthritis or polyarthritis JIA [15], were recruited for this study. Oligoarticular JIA was defined as arthritis affecting 1 to 4 joints during the first 6 months of disease, and polyarthritis as arthritis affecting 5 or more joints during the first 6 months of disease with a negative test for rheumatoid factor. Moderate disease activity was assessed in all patients based on the Juvenile Arthritis Disease Activity Score 27 (JADAS-27). The JADAS-27 (range 0–57) was calculated by summing the scores of four criteria: physician’s global assessment of disease activity (PGA) on a 10 cm visual analogue scale (VAS); parent/patient global assessment of well-being on a 10 cm VAS; active arthritis, defined as joint swelling or limitation of movement accompanied by pain and tenderness, assessed in 27 joints; and erythrocyte sedimentation rate (ESR). Moreover, the accuracy of diagnosis was confirmed by laboratory tests, namely, determining indicators of inflammatory responses, i.e., erythrocyte sedimentation rate (ESR, Westergren technique) and C-reactive protein (CRP, immunonephelometric assay), measuring rheumatoid factor (RF, latex-enhanced immunoturbidimetric test), and by determining the titer of antinuclear antibodies (ANA, indirect immunofluorescence assay). All methods were carried out in accordance with relevant guidelines and regulations. Baseline characteristics of patients are presented in Table 1. The exclusion criteria included other forms of JIA according to the ILAR criteria, as well as any other chronic or autoimmune diseases, traumas, or surgical procedures affecting the locomotor system, renal or liver diseases, withdrawal from therapy during the study period, or ineffectiveness of the therapy.

The first blood samples (for the quantitative assessments of CS/DS, IGF-1, and IGFBP-3) were taken from children with JIA before they were treated with conventional synthetic disease-modifying anti-rheumatic drugs. In turn, second blood samples were taken from the same patients after therapy, when the clinical outcomes had improved, i.e., an average of 11.60 ± 0.21 months after the beginning of the therapy. Clinical improvement was determined using the criteria of the ACR Pediatric 30. Until diagnosis was made, symptomatic treatment with non-steroidal anti-inflammatory drugs was used (ibuprofen and naproxen were the most commonly used agents). Treatment with oral glucocorticoids (at a maximal dose of 1 mg of a prednisone equivalent per kilogram per day, with gradual dose reduction), sulfasalazine (30 mg per square meter of body-surface area), and methotrexate (≤15 mg per square meter of body-surface area once a week) was prescribed in all patients with JIA. 

The reference material comprised blood samples collected from forty-five healthy children (thirty-three girls, thirteen boys), at age-matched to the JIA patients, i.e., before puberty, assessed according to the Tanner scale. The healthy children included in our study did not have any inflammatory or other diseases that required hospitalization, and had not undergone surgical procedures during the previous year. What is more, they had not received pharmacological treatment recently before the study, and the results of routine laboratory tests were normal for their age group. The clinical data of the healthy individuals and JIA patients enrolled in our study are shown in Table 1.

All subjects gave their informed consent for inclusion before they participated in the study. The study was conducted in accordance with the Declaration of Helsinki, and the protocol was approved by the Local Bioethics Committee of the Medical University of Silesia in Katowice (L.dz.KNW/002/KB1/142/I/09, KNW/0022/KB/94/19). 

No conflicts of interest occurred during implementation and completion of this study.

### 2.2. Biochemical Analysis

Venous blood samples were collected after overnight fasting and placed into tubes without anticoagulant. After centrifugation, the serum samples obtained both from healthy individuals and JIA patients were divided into portions and stored at −80 °C till the initiation of the study.

#### 2.2.1. The Assay of the Concentration of CS/DS

The quantitative evaluation of CS/DS in the blood serum of healthy children and JIA patients was performed by assessing the levels of unsaturated disaccharide subunits resulting from enzymatic digestion of isolated GAGs with the use of chondroitinase ABC, i.e., an enzyme that specifically depolymerizes CS/DS.

Hence, in the first stage of the assay, GAGs were isolated from the serum samples using multistage enzymatic and alkaline hydrolysis processes. Then, complexation reaction of these compounds with cetylpyridinium chloride took place according to the method described by us previously [12]. The total quantity of GAGs, including CS/DS, was determined via hexuronic acid assay according to the Blumenkrantz and Asboe-Hansen [16] method, with analytical sensitivity of 0.5 mg/L and calibration ranging from 0.5 to 50 mg/L. The coefficient of intra-assay variation was less than 6%.

In the next step, the isolated GAGs were depolymerized with chondroitinase ABC. Chondroitinase ABC [E.C.4.2.2.4] is a bacterial lyase that breaks the β (1 → 4) glycosidic bond between N-acetylgalactosamine and hexuronic acid (Hex-A) residues within galactosaminoglycans (GalNAc). The products of the effect of this bacterial lyase on the CS/DS chains are unsaturated disaccharides of the formula ΔHex-A-GalNAc, with a double bond between the fourth and fifth carbon atoms of the acid residue and six-sugar fragments of the chains of chondroitin sulphates and dermatan sulphates, which are resistant to chondroitinase ABC.

Degradation of CS/DS chains in 4 µg of hexuronic acids with chondroitinase ABC (activity of 0.02 IU) was carried out in 50 mM Tris–HCl buffer at pH 8.0 for 30 min at 37 °C. Both before the addition of the enzyme and after the end of digestion, the absorbance of the test samples and the standard dermatan sulfate (DS) solutions was measured at a wavelength of λ = 232 nm using a microplate reader (Infinite M200, Tecan, Männedorf, Switzerland). The concentration of unsaturated disaccharides, derived from digestion of CS/DS from the blood plasma of healthy individuals and JIA patients, was determined from the calibration curve created for standard DS solutions with concentrations of 20, 15, 10, 7.5, 5, 2.5, and 1.25 µg/mL.

#### 2.2.2. The Assay of the Concentration of IGF-1 and IGFBP-3

The IGF-1 and IGFBP-3 levels were measured using blindly tested coded plasma samples, in duplicate. Determination of a single parameter was completed within a day; consequently, the inter-assay variation was insignificant. 

Enzymatic immunoassays (ELISA) were used to quantify the IGF-1 and IGFBP-3, following the manufacturer’s protocol. We used ELISA kits dedicated exclusively to scientific research. Serum concentrations of IGF-1 were determined with an ELISA kit from LDN Labor Diagnostika Nord (Nordhorn, Germany), with a minimum detection of 9.75 ng/mL. Serum concentrations of IGFBP-3 were determined with a ELISA Functional Insulin-like Growth Factor Binding Protein 3 (IGFBP-3) ELISA kit from Mediagnost (Reutlingen, Germany), with a minimum detection of 0.18 ng/mL. 

For all parameters tested, the intra-assay variability was less than 8%.

### 2.3. The Statistical Analysis of the Results

The obtained results were statistically analyzed using Statistica 13.3 software (TIBCO Software Inc., Cracow, Poland). The analysis included the following steps: 1. testing the normality of the distribution for a given trait with the Shapiro–Wilk test; 2. testing the equality of variance with the Snedecor–Fisher test; 3. preparing descriptive characteristics for traits with an abnormal distribution, using the median (Me) as a measure of position, and the interquartile range (lower quartile (Q1), upper quartile (Q3)) as a typical range of variation; 4. testing the significance of differences in the mean values of a given trait for the control group and the study groups for traits with asymmetric distribution with the use of the Mann–Whitney U test (non-parametric alternative to the Student’s *t*-test for independent samples), and for traits with asymmetric distribution with the use of a pairwise Wilcoxon test (non-parametric alternative to the Student’s *t*-test for related variables) in both groups of individuals studied. Additionally, analysis of the receiver operating characteristic (ROC) curve was conducted to evaluate the diagnostic utility of the analyzed biomarkers. Pearson’s correlation coefficient modified by Bonferroni’s multivariate correction was used for statistical analysis of correlations between variables. For all the tests and statistical analyses, the significance level was *p* < 0.05.

## 3. Results

The results of CS/DS, IGF-1, and IGFBP-3 concentration measurements in the serum of all examined groups, i.e., the group of healthy children (the control group), the group of newly diagnosed, untreated JIA children, and the group of the same children after their treatment and clinical improvement, broken down by sex and JIA type, are presented in Table 2, Table 3 and Table 4.

### 3.1. Serum Levels of CS/DS in Healthy Children and JIA Patients

Based on the obtained results, we found a significant (*p* = 0.00004) reduction in the serum concentrations of CS/DS in patients with untreated JIA, compared with the controls (Table 2). Moreover, the therapy to ameliorate inflammation that was administered in JIA patients resulted in a significant (*p* = 0.000003) increase in the serum levels of CS/DS in these patients (Table 2). However, treated JIA patients still had markedly (*p* = 0.026) lower serum levels of CS/DS than the control subjects (Table 2). 

Changes in the CS/DS levels in the blood of afflicted girls compared with those characterizing boys showed similar tendencies in the course of JIA, but with a different intensity. Consequently, in the blood of girls with untreated arthropathy, CS/DS was found at significantly (*p* = 0.00007) lower concentrations compared with the controls. The administered treatment contributed to a significant (*p* = 0.00013) increase in the serum levels of the tested matrix components. The CS/DS level in treated girls with arthropathy did not differ (*p* > 0.05) from the level of glycans in the control group. Boys with untreated arthropathy showed a significant (*p* = 0.010) decrease in CS/DS levels compared with healthy boys. Moreover, similarly to the girls’ results, it was found that the therapy led to a significant (*p* = 0.0033) increase in the concentration of glycans in the blood of the patients. The blood levels of CS/DS in boys with stabilized clinical status were comparable (*p* > 0.05) to those in the controls (Table 2).

Furthermore, the levels of CS/DS in the blood of children with newly diagnosed JIA, both oligoarticular (*p* = 0.0003) and polyarticular (*p* = 0.00009) forms, were significantly lower compared with the healthy children (Table 2). However, the data obtained from children with stabilized clinical status indicated that the impact of disease-modifying therapy on the level of circulating CS/DS depended on the form of JIA. Although treatment of patients with oligoarticular and polyarticular JIA led to significant increases (*p* = 0.0003, *p* = 0.0015, respectively) in CS/DS levels compared with pretreatment status levels before therapy, we observed normalization of this parameter only in the patients with oligoarticular forms of the disease (Table 2). In the treated patients with polyarticular JIA, the CS/DS level was still significantly lower (*p* = 0.047) compared with the controls. Moreover, the CS/DS concentration in the blood of untreated patients with polyarthritis was significantly lower (*p* = 0.03) than in patients with oligoarthritis. However, a similar difference was not found in the patients after treatment (Table 2).

### 3.2. Serum Levels of IGF-1 in Healthy Children and JIA Patients

A significant (*p* = 0.0044) reduction in the IGF-1 level in the serum of children with untreated JIA in comparison with the controls was demonstrated. Moreover, it was found that the treatment contributed to an increase (*p* = 0.00009) in the concentration of the assessed factor in the blood of afflicted individuals, to a level typical of healthy children. The IGF-1 level in treated patients did not differ (*p* > 0.05) from the level in the control group (Table 3). 

It was also found that the concentrations of IGF-1 in the blood of untreated girls with JIA did not differ (*p* > 0.05) from the concentrations in the group of healthy girls. It was shown that the applied therapy contributed to a significant increase in IGF-1 serum levels in treated girls, in relation to both the blood of the same children before treatment (*p* = 0.0015) and that of healthy children (*p* = 0.011). However, in the blood of boys with untreated arthropathy, the level of IGF-1 was significantly (*p* = 0.0043) lower than in healthy boys. Moreover, it was shown that the treatment contributed to the normalization of concentrations of the assessed growth factor in the blood of boys with JIA. The demonstrated level of IGF-1 in the blood of treated male patients did not differ (*p* > 0.05) from that of healthy boys, but it was significantly (*p* = 0.021) higher than the concentration observed before the therapy (Table 3).

Moreover, compared with the controls, a significant (*p* = 0.0007) reduction in the concentration of IGF-1 in the serum of children with diagnosed oligoarticular JIA was demonstrated. The therapy that led to clinical improvement in these patients simultaneously contributed to a significant (*p* = 0.0016) increase in serum levels of the assessed factor. IGF-1 levels in treated patients with oligoarticular JIA did not differ (*p* > 0.05) compared with the control group. On the other hand, IGF-1 serum levels in untreated patients with polyarticular forms of the disease did not differ (*p* > 0.05) from the concentration shown in the control group. However, the treatment of these patients led to significant increases in the serum level of the tested compound in comparison to the concentration of IGF-1, both in the blood of the same children before treatment (*p* = 0.019) and that of healthy children (*p* = 0.0087) (Table 3). 

### 3.3. Serum Levels of IGBP-3 in Healthy Children and JIA Patients

There was a significant (*p* = 0.0002) decrease in the IGFBP-3 level in the serum of children with untreated JIA, in comparison to the controls. It was also shown that the treatment contributed to a significant (*p* = 0.0006) increase in the concentration of IGFBP-3 in the blood of patients compared with the pre-treatment situation. Moreover, the concentration of the assessed binding protein in the blood of the treated patients was comparable (*p* > 0.05) to the control group (Table 4).

Changes in IGFBP-3 levels in the blood of afflicted girls compared with those characteristic of the boys revealed different tendencies in the courses of JIA. In the blood of girls with newly diagnosed arthropathy, the concentration of the assessed protein did not differ (*p* > 0.05) from the concentration in the group of healthy girls. On the other hand, the improvement in the clinical status of the afflicted girls was accompanied by a significant (*p* = 0.0043) increase in the IGFBP-3 serum level. The concentration of the assessed binding protein in the blood of the treated girls did not differ (*p* > 0.05) from the control, whereas, in the blood of boys with untreated arthropathy and in the same patients after the treatment, the concentration of the assessed compound was significantly (*p* = 0.0008, *p* = 0.022, respectively) lower than in the blood of healthy boys. What is more, the level of IGFBP-3 in the blood of the treated boys with JIA was significantly higher (*p* = 0.026) in comparison to its level determined in these patients before the therapy (Table 4).

It was also found that the blood level of IGFBP-3 in children with newly diagnosed, oligoarticular forms of JIA was significantly (*p* = 0.00005) lower in comparison to the controls. It was also shown that the treatment of these patients led to a significant (*p* = 0.0042) increase in the serum level of the assayed compound. The concentration of IGFBP-3 in the blood of treated patients with oligoarticular forms of JIA did not differ (*p* > 0.05) from the concentration in the control group. However, serum levels of IGFBP-3 characterizing patients with a polyarticular type of the disease, both before and after treatment, were comparable with each other (*p* > 0.05) and with the concentration characteristic of healthy children (*p* > 0.05) (Table 4).

### 3.4. Usefulness of Levels of the Circulating CS/DS, IGF-1, and IGFBP-3 in the Diagnosis of JIA

To assess the diagnostic utility of the examined analytes, receiver operating characteristic (ROC) curve analysis was conducted; the obtained results are presented in Figure 1 and Table 5. The optimal cut-off point for each of these parameters was determined by the Youden index. As can be seen from Table 5, the blood concentration of CS/DS has excellent power to distinguish JIA patients from healthy controls, based on the very high value of the area under the ROC curve (AUC) (0.947). In contrast, the circulating IGF-1 and IGFBP-3 are both characterized by similar, relatively good abilities to differentiate patients from control individuals, as judged from the moderately low AUC values for these analytes (0.724 and 0.761, respectively). The other indicators of diagnostic accuracy for the examined parameters were determined at the optimal cut-off point for each of these analytes. The obtained results indicate that the blood concentration of CS/DS has especially high diagnostic potential for recognition of JIA patients and healthy subjects, resulting from high sensitivity and specificity values. Furthermore, the power of positive and especially, negative results of this analyte to confirm or exclude the disease is significant, based on the obtained PPV and NPV values. In turn, at their optimal cut-off points, the blood levels of IGF-1 or IGFBP-3 seem to be more useful for recognizing JIA patients than healthy subjects. Moreover, as indicated by the predictive values, especially the negative results of these parameters, they were able to exclude JIA with high probability (Table 5). 

### 3.5. The Relationship of CS/DS with IGF-1 and IGFBP-3

To investigate the mutual connections between metabolism of CS/DS and IGF-1, which is an important stimulator of ECM component biosynthesis, we assessed the relationships between circulating fractions of these parameters in the JIA-affected persons and healthy individuals. Moreover, we also estimated the correlations between the serum concentrations of CS/DS and IGFBP-3 in these people, these being significant regulators of IGF-1 activity. This analysis of the serum levels of CS/DS and IGF-1 (Figure 2a) elicited the existence of significant negative relationships between these parameters in the group of children with treated JIA (r = −0.2929, *p* = 0.05) and especially, in untreated patients with this disease (r = −0.580, *p* < 0.0001). Moreover, only in these individuals was a significant (r = −0.506, *p* = 0.0001) negative correlation between circulating CS/DS and serum concentration of IGFBP-3 also found (Figure 2b). 

### 3.6. The Relationship of CS/DS, IGF-1, IGFBP-3 with CRP and ESR

In order to assess the involvement of CS/DS, IGF-1, and IGFBP-3 metabolism in the inflammation, we estimated the relationships between circulating levels of these parameters and inflammatory markers i.e., CRP and ESR, in patients with JIA, both before treatment and after clinical improvement resulting from the applied therapy (Table 6). In children with newly diagnosed JIA we observed a significant positive correlation (r = 0.601, *p* = 0.0005) only between the serum level of CS/DS and the level of CRP in these patients. In contrast, no significant correlations between the CS/DS concentration and the selected inflammatory biomarkers were demonstrated in the group of treated JIA patients. Furthermore, no significant correlation was found between serum IGF-1 levels and the values of either CRP or ESR in the untreated and treated JIA patients. In turn, levels of circulating IGFBP-3 demonstrated a significant negative relationship (r = −0.423, *p* = 0.019) only with ESR value, solely in children with untreated JIA (Table 6).

## 4. Discussion

Our results clearly indicate that significant alterations occur in the levels of circulating CS/DS, IGF-1, and IGFBP-3 in children with untreated JIA, and these changes are sensitive to anti-inflammatory treatment. These observations suggest that JIA is associated with metabolic remodeling that has systemic manifestations. Moreover, the level of circulating CS/DS in particular has great potential utility in the diagnosis of JIA, as evidenced by the ROC curve analysis we performed. 

Many reports have shown that both in health and in various diseases, the blood fraction of CS/DS consists of two components: CS as a predominant component and DS as a minor, if not negligible, one [17,18,19,20]. Thus, the nearly two-fold reduction in the serum concentration of CS/DS in untreated JIA patients compared with healthy controls may have resulted, at least in part, from alterations in CS levels. It is commonly accepted that the blood fraction of this glycan originates from two main sources: (1) the hepatic production and secretion of members of the inter-α-inhibitor (IαI) family and (2) the influx of CS-attaching proteoglycan (CSPGs) degradation products from tissues [20]. Interestingly, under inflammatory conditions, such as in untreated JIA patients, the expression of certain IαI members is markedly diminished and the secreted molecules are quickly consumed, due to their participation in stabilizing the hyaluronan network or inhibiting various serine proteases derived from activated leucocytes [20,21]. On the other hand, cartilage is a significant supplier of CSPG degradation products to the blood; this includes articular cartilage, which contains large amounts of glycoproteins such as aggrecan, decorin, and biglycan in its ECM [22,23]. 

However, the metabolism of CSPGs undergoes marked alterations in the articular cartilage of JIA patients, as evidenced by the significant reductions in concentration of both sulfated glycosaminoglycans and aggrecan degradation products in the synovial fluid of these individuals, compared not only with osteoarthritis patients but also healthy subjects [24]. These results suggest that in JIA patients, articular cartilage might “deliver” less CS-containing species to the blood. Thus, it is possible that both of the processes described above could contribute to the significant decrease in circulating CS/DS levels observed in untreated JIA patients. Interestingly, we found some differences in this analyte not only between various forms of JIA but also between boys and girls with the disease; this observation is in line with findings reported by Shevchenko et al. [25]. Thus, gender is an additional factor affecting blood levels of CS/DS in JIA patients, apart from the patient’s age at the time of diagnosis and the duration of the disease [25].

Inflammation is closely related to the increased degradation of various molecules, including CSPGs, in affected tissues. The extracellular degradation of these glycoproteins is catalyzed by several enzymes, including some proteases such as aggrecanases from the ADAMTS family, or matrix metalloproteinases (MMPs), as well as selected hyaluronidases [3,25,26,27]. In addition, the GAG part of PGs can be also partly depolymerized by free radicals, whose production is markedly increased under inflammatory conditions [3]. Notably, increased activity of several aggrecanases and MMPs has been found in both the synovial fluid and serum of JIA patients [11,12,27,28,29,30,31,32,33], indicating enhanced degradation of ECM components, including CSPGs, in the articular cartilage of the affected joints. Interestingly, other pathological processes involving articular cartilage-located inflammation, such as osteoarthritis or rheumatoid arthritis, are associated with significant increases in levels of circulating glycosaminoglycans, including CS/DS [18,19]. We did not observe the same effect in untreated JIA patients, even though the serum level of CS/DS was strongly positively correlated with CRP concentration in these individuals. Thus, it is tempting to speculate that the degradation of CSPGs in the affected joints of untreated JIA patients has unique features distinguishing it from other inflammatory diseases involving articular cartilage. This hypothesis is also supported by the results of profile analysis of aggrecan degradation products in the synovial fluid of affected subjects [24]. This distinct characteristic might be reflected in the generation of specific CSPG degradation products that either are processed rapidly and almost completely locally or do not accumulate in the blood, due to quick clearance. However, it cannot be ruled out that the reduced levels of circulating CS/DS in untreated JIA patients result from a specific pathomechanism of the disease, involving exhaustive articular cartilage degradation and occurring in the early preclinical stages. Thus, when disease symptoms appear, the cartilage pool of CSPGS is markedly depleted, also due to disturbed biosynthesis of these molecules. Nevertheless, the observed differences in the levels of circulating CS/DS between patients with JIA and those with osteoarthritis or rheumatoid arthritis may indicate a possible reason for the different clinical courses of these diseases, as glycans are known modulators of the NFκB pathway, which is a key player in inflammation [34]. Based on the ROC curve analysis, our data clearly demonstrate that circulating CS/DS levels can well discriminate JIA patients from healthy individuals, thus facilitating disease diagnosis. Furthermore, this parameter behaves differently in JIA compared with other cartilage-involving inflammatory diseases such as rheumatoid arthritis or osteoarthritis, as suggested by the literature data mentioned above. Therefore, blood levels of CS/DS might be potentially useful in distinguishing JIA from these diseases. However, it remains unclear whether other common conditions, including those associated with inflammatory response, such as post-traumatic joint injury or intensive exercise, could affect this analyte.

Meanwhile, our data suggest that circulating CS/DS levels could be useful for monitoring the clinical status of treated JIA patients. However, it should be emphasized that our study concerning the utility of CS/DS blood concentration for both JIA diagnosis and monitoring of the efficacy of the disease treatment is preliminary. Additional examinations in larger patient groups are necessary to better assess the diagnostic and therapeutic implications of the parameter, especially in relation to radiological imaging results and clinical manifestation of the disease. Another factor determining the utility of circulating CS/DS is the method of measurement. In our study, we used a complex procedure including the isolation of these compounds and their quantification by hexuronic acid assay. This procedure can be greatly simplified to a direct reaction of chondroitinase ABC on CS/DS in plasma (serum) samples with measurement of absorbance at 232 nm, which corresponds to the maximum light absorbance for unsaturated disaccharides, being products of specific enzymatic degradation of these glycan compounds. However, a full analytical characterization of this simplified method remains to be performed. On the other hand, further research should focus on identifying the structural signature of circulating CS/DS, especially in relation to the sulfation patterns of these compounds, which may be unique for JIA. Such a feature could facilitate the development of an immunoassay, making the analyte more practical for use in laboratory settings.

Articular cartilage homeostasis is controlled by growth factors, including IGF-1. This molecule influences not only chondrocyte functions such as proliferation, differentiation, survival, and hypertrophy but also strongly stimulates the biosynthesis of cartilage ECM components [35,36,37]. These effects are optimized by cooperation of IGF-1 with growth hormone [38]. Circulating IGF-1 is primarily of hepatic origin, but its blood level also depends on age and sex. Moreover, the blood concentration of IGF-1 is regulated by IGFBPs, mainly by variant 3, which forms a complex with the growth factor, prolonging its half-life in the vascular space [37]. Both IGF-1 and IGFBP-3 were assessed in patients with JIA [39,40,41,42,43,44,45]. Our present study clearly showed that circulating IGF-1 and IGFBP-3 levels were markedly reduced in untreated JIA patients, especially in oligoarthritis as well as in boys. These results are in line with report by Lundell [39]. The mechanism underlying this phenomenon may involve increased activity of pro-inflammatory cytokines such as TNF-α, IL-1, and IL-6 [40,41], which have a multidirectional impact on IGF-1 and IGFBP biology. This influence is realized by an inhibitory effect on IGF-1 expression, hepatic secretion, and intracellular signaling. This latter impact occurs via attenuation of IGF-1-induced activation of the MAP/ERK 1/2 and phosphatidylinositol-3 kinase pathways in chondrocytes, leading to resistance of these cells to the growth factor, which interferes with the function of the epiphysial plate in children [42,43,44,45,46,47,48]. Suppression of IGFBP-3 biosynthesis in Kupffer cells and/or stimulation of this protein’s degradation resulted in decreased blood concentration of IGF-1–IGFBP complexes and increased renal clearance of the growth factor [40,41]. 

Thus, it seems that the significant negative correlations we observed between the circulating CS/DS and IGF-1 or IGFBP-3 should rather be considered as resulting from the regulatory impact of the inflammatory environment on the metabolism of each of these molecules. This hypothesis is supported by the finding that such relationships were observed only in untreated JIA patients, in whom high activity of pro-inflammatory cytokines would not only reduce the blood concentrations of both examined proteins but also lead to an increase in circulating CS/DS via stimulation of CSPG degradation in the articular cartilage of the affected joints. Nevertheless, more direct connections between IGF-1 or IGFBP-3 and CS/DS may also exist. The growth factor can modulate the biosynthesis of CSPGs, and this effect may be tissue-specific. In contrast to glomerular cells or peritubular cells of the testis, in which IGF-1 stimulated CSPGs production [47], direct injection of this growth factor into osteoarthritic joints in mice had only a small effect [49]. In turn, IGFBPs can bind to glycosaminoglycans, including CS/DS, in tissues [50]. This interaction reduces the affinity of IGFBPs to IGF-1, triggering the dissociation of complex between these proteins and leading to a short-term increase in the bioavailability of the growth factor alone [37,51]. 

Our results demonstrated that the administration of anti-inflammatory drug into JIA patients, thereby improving their clinical status, led to a significant increase in the level of circulating CS/DS compared with levels before treatment. However, the treatment did not lead to complete normalization of this parameter, especially in girls and patients with polyarthritic forms of the disease, indicating that the regulation of CS/DS metabolism is more complex and extends beyond the alleviation of inflammation. 

## 5. Conclusions

Understanding the changes in CS/DS metabolism in the course of JIA, regulated by various mechanisms and depending on both the disturbed IGF/IGFBP axis and on the hyperactivity of depolymerizing factors, including pro-inflammatory, pro-oxidative, and proteolytic factors, may allow the implementation of new diagnostic tools and therapeutic strategies in children with JIA. We suggest that CS/DS may be a useful biomarker not only for disease diagnosis but also disease activity, and it could be used to assess treatment efficacy towards remission. However, drawing strong conclusions requires a larger sample size and particularly, more balanced representation of male patients. The relatively small number of male patients in this study reflects the low prevalence of JIA among children in Poland (5–6.5 cases per 100,000 children) [52] and the higher incidence of JIA in females. The small sample size in the study is the main limitation of this work.

## Figures and Tables

**Figure 1 biomolecules-14-01526-f001:**
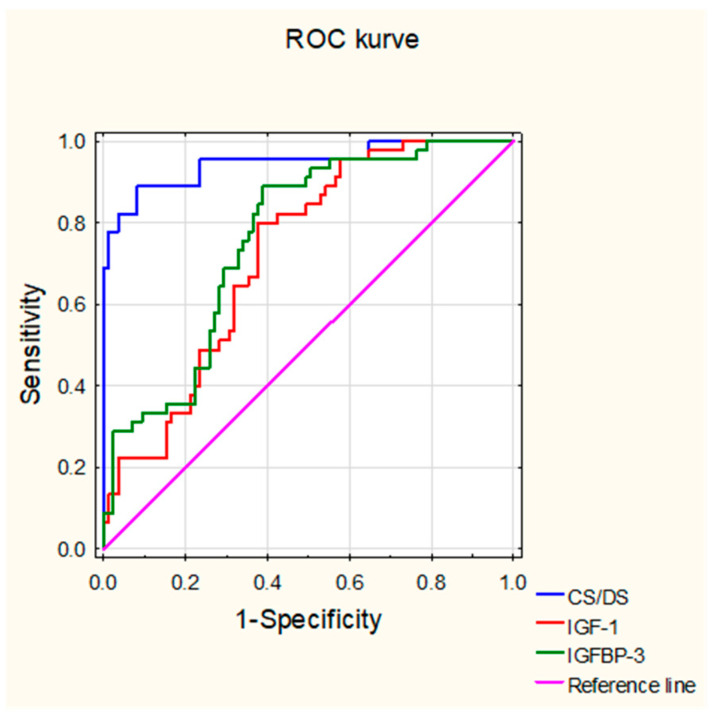
ROC curves for circulating CS/DS, IGF-1, and IGFBP-3 in JIA patients.

**Figure 2 biomolecules-14-01526-f002:**
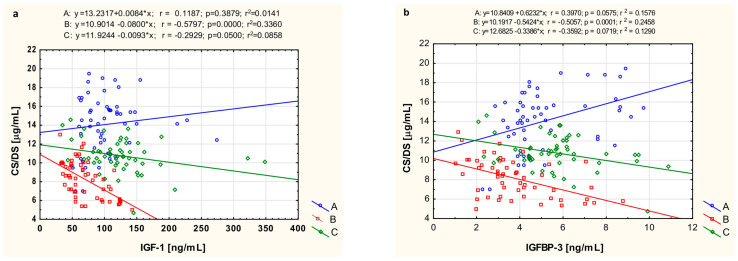
Graphical analysis of the linear relationship between serum concentrations of CS/DS and IGF-1 (**a**) as well as IGFBP-3 (**b**) in healthy children (A, control subjects), untreated JIA patients (B), and the same patients after treatment (C).

**Table 1 biomolecules-14-01526-t001:** Clinical data of control subjects and JIA patients.

Parameter	Control Subjects (*n* = 45)	Untreated JIA Patients (*n* = 55)	JIA Patients After Treatment (*n* = 55)
Age (years)	8.25 ± 4.03	7.31 ± 4.86	8.28 ± 4.71
Sex, female/male	33/12	41/14	41/14
JADAS-27	-	17.50 (15.50–21.50)	1.00 (1.00–1.50) ^b^
WBC (10^3^/μL, by fluorescent flow cytometry method)	5.35 ± 2.19	11.21 ± 2.98 ^a^	7.25 ± 2.57 ^b^
RBC (10^6^/μL, by impedance method)	4.95 ± 0.35	3.85 ± 0.93 ^a^	4.54 ± 0.67
Hb (g/dL, by colorimetric method)	13.85 ± 0.94	11.43 ± 3.02 ^a^	13.06 ± 4.31 ^b^
PLT (10^3^/μL, by impedance method)	283.47 ± 73.83	349.19 ± 58.89	316.91 ± 57.76
GPT (U/L, by spectrophotometric measurement)	18.58 ± 12.82	22.54 ± 11.01	25.52 ± 14.13 ^a,b^
GOT (U/L, by spectrophotometric measurement)	23.70 ± 12.64	25.85 ± 12.47	28.98 ± 15.15 ^a,b^
Creatinine (mg/dL, by Jaffe method)	0.68 ± 0.15	0.65 ± 0.21	0.78 ± 0.43
ESR (mm/h, by Westergren technique)	9.02 ± 3.03	41.47 ± 16.29 ^a^	13.88 ± 11.05 ^b^
CRP (mg/L, by immunonephelometric assay)	0.59 (0.27–1.28)	30.65 (19.15–44.58) ^a^	2.73 (0.29–5.11) ^b^
ANA (by indirect immunofluorescence assay)	-	61% (positive)	61% (positive)
RF (by latex-enhanced immunoturbidimetric test)	-	100% (negative)	100% (negative)

Results are expressed as mean ± SD or median (quartile Q1–quartile Q3); ^a^ *p* < 0.05 compared with the control group; ^b^ *p* < 0.05 compared with untreated JIA patients; JADAS-27, Juvenile Arthritis Disease Activity Score-27; WBC, white blood cell; RBC, red blood cell; Hb, hemoglobin; PLT, platelet; GPT, glutamic pyruvic transferase; GOT, glutamic oxaloacetic transaminase; ESR, erythrocyte sedimentation rate; CRP, C-reactive protein; ANA, antinuclear antibody; RF, rheumatoid factor.

**Table 2 biomolecules-14-01526-t002:** The distribution patterns of plasma CS/DS in the healthy individuals (control subjects) and JIA patients.

	Control Subjects	Untreated JIA Patients	JIA Patients After Treatment
CS/DS [µg/mL]			
Total CS/DS	14.48 (10.23–15.77)	8.26 (6.25–9.66) *	10.68 (10.03–11.73) *^,#^
(N)	(45)	(55)	(55)
Oligoarthritis	14.48 (10.23–15.77)	8.63 (7.36–8.24) *^,$^	11.16 (10.39–12.62) ^#^
(N)	(45)	(31)	(31)
Polyarthritis	14.48 (10.23–15.77)	7.09 (5.63–8.41) *	10.55 (10.27–11.01) *^,#^
(N)	(45)	(24)	(24)
Girls	14.09 (9.66–14.89)	7.21 (5.75–8.94) *	10.86 (10.29–12.65) ^#^
(N)	(33)	(41)	(41)
Boys	12.99 (12.02–15.77)	8.57 (8.13–9.78) *	12.55 (10.03–13.68) ^#^
(N)	(12)	(14)	(14)

Results are expressed as medians (quartile Q1–quartile Q3), CS/DS, chondroitin sulfate/dermatan sulfate; * *p* < 0.05 compared with the control group; ^#^
*p* < 0.05 compared with untreated JIA patients, ^$^ *p* < 0.05 compared with untreated polyarthritis JIA patients.

**Table 3 biomolecules-14-01526-t003:** The distribution patterns of plasma IGF-1 in the healthy individuals (control subjects) and JIA patients.

	Control Subjects	Untreated JIA Patients	JIA Patients After Treatment
IGF-1 [ng/mL]			
Total IGF-1	96.92 (76.04–128.59)	66.04 (49.45–96.80) *	120.60 (92.78–138.77) ^#^
(N)	(45)	(55)	(55)
Oligoarthritis	96.92 (76.04–128.59)	57.13 (45.35–84.06) *	95.18 (65.01–120.32) ^#^
(N)	(45)	(31)	(31)
Polyarthritis	96.92 (76.04–128.59)	111.26 (94.83–178.35)	143.75 (90.35–167.61) *^,#^
(N)	(45)	(24)	(24)
Girls	90.94 (73.69–137.09)	73.42 (54.78–115.74)	121.16 (88.31–150.99) *^,#^
(N)	(33)	(41)	(41)
Boys	118.47 (100.08–169.34)	69.67 (48.23–98.21) *	110.15 (63.93–154.06) ^#^
(N)	(12)	(14)	(14)

Results are expressed as medians (quartile Q1–quartile Q3). IGF-1, insulin-like growth factor 1; * *p* < 0.05 compared to control group; ^#^
*p* < 0.05 compared to untreated JIA patients.

**Table 4 biomolecules-14-01526-t004:** The distribution patterns of plasma IGFBP-3 in the healthy individuals (control subjects) and JIA patients.

	Control Subjects	Untreated JIA Patients	JIA Patients After Treatment
IGFBP-3 [ng/mL]			
Total IGFBP-3	4.84 (4.21–7.75)	3.37 (2.65–4.88) *	5.67 (4.51–6.19) ^#^
(N)	(45)	(55)	(55)
Oligoarthritis	4.84 (4.21–7.75)	3.01 (2.15–4.26) *	5.58 (4.02–6.17) ^#^
(N)	(45)	(31)	(31)
Polyarthritis	4.84 (4.21–7.75)	4.26 (3.84–5.91)	5.69 (4.91–7.35)
(N)	(45)	(24)	(24)
Girls	4.69 (4.10–7.07)	3.68 (2.71–6.75)	5.71 (4.86–7.01) ^#^
(N)	(33)	(41)	(41)
Boys	7.25 (5.28–7.94)	3.07 (2.83–4.28) *	4.55 (3.03–5.46) *^,#^
(N)	(12)	(14)	(14)

Results are expressed as medians (quartile Q1–quartile Q3). IGFBP-3, insulin-like growth factor-binding protein-3; * *p* < 0.05 compared to control group; ^#^
*p* < 0.05 compared to untreated JIA patients.

**Table 5 biomolecules-14-01526-t005:** The results of ROC curve analysis of biomarkers CS/DS, IGF-1, and IGFBP-3.

	CS/DS	IGF-1	IGFBP-3
AUC	0.947	0.724	0.761
Opitmal cut-off	10.374 µg/mL	75.411 ng/mL	3.925 ng/mL
Youden Index	0.807	0.424	0.501
Accuracy	0.908	0.685	0.708
Sensitivity	0.889	0.800	0.889
Specificity	0.918	0.624	0.612
PPV	0.851	0.529	0.548
NPV	0.940	0.855	0.912

AUC, the area under the ROC curve; JIA, juvenile idiopathic arthritis; NPV, negative predictive value; PPV, positive predictive value; CS/DS, chondroitin/dermatan sulfate; IGF-1, insulin-like growth factor 1; IGFBP-3, insulin-like growth factor-binding protein 3.

**Table 6 biomolecules-14-01526-t006:** Correlation analysis between serum CS/DS, IGF-1, and IGFBP-3 and CRP, ESR levels in juvenile idiopathic arthritis (JIA) patients.

Parameter	CS/DS r (*p*)	IGF-1 r (*p*)	IGFBP-3 r (*p*)
CRP			
Untreated JIA patients			
	0.601 (0.0005)	−0.233 (0.215)	−0.228 (0.225)
JIA’ patients after treatment			
	−0.318 (0.099)	−0.117 (0.368)	−0. 002(0.990)
ESR			
Untreated JIA’ patients			
	0.283 (0.130)	−0.055 (0.774)	−0.423 (0.019)
JIA’ patients after treatment			
	−0.216 (0.252)	−0.170 (0.370)	0.288 (0.123)

Results are expressed as Pearson correlation coefficients; CS/DS, chondroitin/dermatan sulfate; IGF-1, insulin-like growth factor 1; IGFBP-3, insulin-like growth factor-binding protein 3; CRP, C-reactive protein; ESR, erythrocyte sedimentation rate.

## Data Availability

The datasets analyzed or generated during this study are available from the authors: winsz@sum.edu.pl; kkuznik@sum.edu.pl.

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
