# Peer review of "Serum CS/DS, IGF-1, and IGFBP-3 as Biomarkers of Cartilage Remodeling in Juvenile Idiopathic Arthritis: Diagnostic and Therapeutic Implications"

_biomolecules, 2024, doi:10.3390/biom14121526_

Round 1

Reviewer 1 Report

Comments and Suggestions for Authors

The authors have examined the possibility that serum levels of so-called GAAGs are biomarkers for children with juvenile idiopathic arthritis (JIA).  The metabolism of cartilage and is components, which include proteoglycans and their associated glycosaminoglycan (GAG) side chains is important, especially in JIA.  The authors studied materials in serum samples from 55 Polish Caucasian children of both sexes (forty-one girls, fourteen boys), and accessed also the influence of therapy with  IGF-1.  The authors conclude that the patients had low serum GAAG which increased upon treatment.  They suggest that GAAG levels might be a useful biomarker of disease activity.  Overall, the study is well-written,  interesting and raises several issues in regard to diagnosis and treatment biomarkers for JIA.  However, there are several issues the authors should address. 

1.     The method is to treat serum with chondroitinase ABC, i.e. an enzyme that specifically depolymerizes GAAGs, and measure the level of subsequent cetylpyridinium chloride derivatives using a hexuronic acid assay.  The sensitivity of this assay appears reasonable, but the evidence that only CS/DS fragments are being analyzed, is simply assumed based on using chondroitinase ABC.  Did the authors analyze the fragments by MS or other methods or compositional methods to confirm this? 

2.     The authors state that they:  “….assessed the influence of insulin-like growth factor-1 (IGF-1)/ insulin-like growth factor-binding protein-3 (IGFBP-3) axis (using the ELISA methods) on GAAGs levels in JIA patients, both before and after treatment.”  The did not assess the influence of these but instead measured their levels upon treatment.  The authors need to be clearer about the treatments of these patients, as they state “The treatment with stable doses of non-steroidal anti-inflammatory drugs (ibuprofen and naproxen were the most commonly used agents), oral glucocorticoids (at 127 a maximal dose of 1 mg of a prednisone equivalent per kilogram per day, with gradual dose reduction), sulfasalazine (30 mg per square meter of the body-surface area) and methotrexate (≤15 mg per square meter of the body-surface area once a week), was prescribed.”  Thus, it appears that a variety of treatments were given and these variables are not taken into account. 

3.     Control issues:  The authors compare their results to healthy controls, choosing 45 healthy children (thirty-three girls, thirteen boys), at the age matching the JIA patients.  However, controls in which children might have inflammatory disease is unclear, are not considered, and treatment regimens are also obviously limited to patients. 

4.     The authors found that clinical treatment causes elevations of CS/DS (GAAGs), IGF-1, IGFBP-3 and C-reactive protein.  It is not clear however, in the manuscript, whether the changes in IGF-1, IGFBP-3 and C-reactive protein have already been noted in prior studies. 

5.     It is a bit unclear about what the authors are interpreting to be a ‘diagnostic’ application of their findings, as JIA is readily diagnosed already.  In addition, they did not address whether their finding would aid in a ‘differential diagnoses’ where JIA is only one of the potential clinical causes of the inflammation in these patients.  It would seem, instead, that the authors are suggesting that measuring GAAG levels would aid in monitoring treatment.  However, measurements of IGF-1, IGFBP-3 and C-reactive protein, which are easily made, might be more practical.

6.     The collection of serum after treatment, i.e., the time course, of response, is not really discussed, and it is a bit unclear, but apparently the authors used a single time point of serum collection for their measurements following treatment.  This should be clarified.

There are just a few grammatical issues: 

In the Abstract “…indicate the need for continued to continue therapy to protect…“

Abbreviations:  The use of the abbreviation GAAG is somewhat problematic, as the accepted abbreviation for glycosaminoglycans is GAG, thus this could add confusion.  There seems no need to change this.  What they define as a subset of GAGs, the GAAGs (galactosaminoglycans) are chondroitin sulfate and dermatan sulfate.  Perhaps the better choice would be CS/DS, which researchers in the field readily identify. 

Author Response

Answer to the Reviewer's comment:

We would like to thank the Reviewer for the  evaluation of our article. We are grateful for the important comments, which we fully addressed in the revised manuscript. The text of our manuscript (in each individual part, changes made in blue) has been modified, so as to facilitate its understanding and make it acceptable for publication.

Detailed modifications are presented below:

  1. Comments:

The method is to treat serum with chondroitinase ABC, i.e. an enzyme that specifically depolymerizes GAAGs, and measure the level of subsequent cetylpyridinium chloride derivatives using a hexuronic acid assay.  The sensitivity of this assay appears reasonable, but the evidence that only CS/DS fragments are being analyzed, is simply assumed based on using chondroitinase ABC.  Did the authors analyze the fragments by MS or other methods or compositional methods to confirm this?

Response: Thank you very much for your valuable comment, and at the same time kindly explain. It is well documented (evidenced) that the chondroitinase ABC treatment at pH 8.0, which  we used, specifically degrades only CS/DS (GAAGs), generating unsaturated disaccharides as products when the reaction is complete. However, we agree with the Reviewer’ s suggestion that in the future it would be worthwhile to examine the products of the complete or partial degradation of the plasma/serum CS/DS from JIA patients, especially with respect to the sulfation pattern of these compounds. Such a study could potentially reveal a specific CS/DS sequence as the disease signature  allowing the development of an immunological test to identify patients with JIA. We have added this explanation  to the Discussion section in the revised manuscript [p.13, lines 520-537].

  1. Comments:

The authors state that they:  “….assessed the influence of insulin-like growth factor-1 (IGF-1)/ insulin-like growth factor-binding protein-3 (IGFBP-3) axis (using the ELISA methods) on GAAGs levels in JIA patients, both before and after treatment.”  The did not assess the influence of these but instead measured their levels upon treatment.  The authors need to be clearer about the treatments of these patients, as they state “The treatment with stable doses of non-steroidal anti-inflammatory drugs (ibuprofen and naproxen were the most commonly used agents), oral glucocorticoids (at 127 a maximal dose of 1 mg of a prednisone equivalent per kilogram per day, with gradual dose reduction), sulfasalazine (30 mg per square meter of the body-surface area) and methotrexate (≤15 mg per square meter of the body-surface area once a week), was prescribed.”  Thus, it appears that a variety of treatments were given and these variables are not taken into account.

Response:  Thank you very much for your valuable comment, and at the same time kindly explain. We did not assess the influence of the IGF-1/IGFBP-3 axis on SC/DS levels directly but rather analysed their circulating levels with CS/DS concentrations in JIA patients before and after treatment. We have corrected this descriptions in the revised Abstract of the manuscript to reflect the actual scope of our analysis [p.1, lines 14-19].

Moreover,  in order to standardise the group of patients, we included children treated with a combination of oral glucocorticoids, sulfasalazine, and methotrexate. As suggested by the Reviewer, we have modified the description of the treatment of JIA patients in the Materials and Methods section [p.3, lines 131-142].

  1. Comments:

Control issues:  The authors compare their results to healthy controls, choosing 45 healthy children (thirty-three girls, thirteen boys), at the age matching the JIA patients.  However, controls in which children might have inflammatory disease is unclear, are not considered, and treatment regimens are also obviously limited to patients.

Response: In accordance with the reviewer's comment, we have clarified the description of the control group in the Materials and Methods section. We indicated that children with any symptoms of inflammation were excluded from our study to ensure the validity of comparisons [p.3/4, lines 145-147].

  1. Comments:

The authors found that clinical treatment causes elevations of CS/DS (GAAGs), IGF-1, IGFBP-3 and C-reactive protein.  It is not clear however, in the manuscript, whether the changes in IGF-1, IGFBP-3 and C-reactive protein have already been noted in prior studies.

Response: In accordance with a reviewer's comment, we have added information about studies on IGF-1, IGFBP-3 and CRP blood levels in JIA patients, in the Discussion section [p.13, lines 545-546; References has been updated].

  1. Comments:

It is a bit unclear about what the authors are interpreting to be a ‘diagnostic’ application of their findings, as JIA is readily diagnosed already.  In addition, they did not address whether their finding would aid in a ‘differential diagnoses’ where JIA is only one of the potential clinical causes of the inflammation in these patients.  It would seem, instead, that the authors are suggesting that measuring GAAG levels would aid in monitoring treatment.  However, measurements of IGF-1, IGFBP-3 and C-reactive protein, which are easily made, might be more practical.

Response: Given the Reviewer’s comments, in the revised manuscript we have modified (added) the text fragment (in the Discussion section) regarding the diagnostic usefulness of the circulating GAAG level in JIA patients [p.13, lines 511-537].

  1. Comments:

The collection of serum after treatment, i.e., the time course, of response, is not really discussed, and it is a bit unclear, but apparently the authors used a single time point of serum collection for their measurements following treatment.  This should be clarified.

Response: Following the Reviewer’s recommendation, we have modified the protocol for blood sampling in the Materials and Methods section [p.3, lines 131-142].

  1. Comments:

In the Abstract “…indicate the need for continued to continue therapy to protect…“

Response: We apologise for the grammatical errors, which have been corrected in the Abstract section of the article.

  1. Comments:

Abbreviations:  The use of the abbreviation GAAG is somewhat problematic, as the accepted abbreviation for glycosaminoglycans is GAG, thus this could add confusion.  There seems no need to change this.  What they define as a subset of GAGs, the GAAGs (galactosaminoglycans) are chondroitin sulfate and dermatan sulfate.  Perhaps the better choice would be CS/DS, which researchers in the field readily identify. 

Response: In line with the reviewer's comment, we have changed the abbreviation GAAGs to CS/DS,  to make the work easier to understand.

Reviewer 2 Report

Comments and Suggestions for Authors

The authors reported that changes of GAAGs during the course of JIA, influenced by both impairment of the IGF/IGFBP axis and inflammation, indicate the need for continued to continue therapy to protect patient from potential disability. The authors  suggest that GAAGs may be a useful biomarker of disease activity and could be employed to assess the treatment efficacy and progress toward remission. 

The paper is almost good, however it will be better if the authors discuss additionally the following points. 

Minor suggestion.

In Table 3,  lines 297-299, the authors wrote “IGF-1 serum levels in untreated patients with polyarticular form of the disease did not differ (p>0.05) from the concentration shown in the control group.” However, the IGF1 levels of 111.26 ng/mL in untreated polyarthritis are higher than that of control, even though without significant difference. Moreover, the IGF1 levels of untreated polyarthritis are higher than that of untreated oligoarthritis. In other untreated JIA patients, IGF1 levels are low, why only in polyarthritis JIA patients IGF1 levels are high ?

In Table 4, the IGFBP3 levels of untreated polyarthritis are higher than that of untreated oligoarthritis. Why IGFBP3 levels are higher in polyarthritis than in oligoartiritis ?

Author Response

Answer to the Reviewer's comment:

We would like to thank the Reviewer for the  evaluation of our article. We are grateful for the important comments, which we fully addressed in the revised manuscript. The text of our manuscript (in each individual part, changes made in blue) has been modified, so as to facilitate its understanding and make it acceptable for publication.

Detailed modifications are presented below:

  1. Comments:

In Table 3,  lines 297-299, the authors wrote “IGF-1 serum levels in untreated patients with polyarticular form of the disease did not differ (p>0.05) from the concentration shown in the control group.” However, the IGF1 levels of 111.26 ng/mL in untreated polyarthritis are higher than that of control, even though without significant difference. Moreover, the IGF1 levels of untreated polyarthritis are higher than that of untreated oligoarthritis. In other untreated JIA patients, IGF1 levels are low, why only in polyarthritis JIA patients IGF1 levels are high ?

  1. Comments: In Table 4, the IGFBP3 levels of untreated polyarthritis are higher than that of untreated oligoarthritis. Why IGFBP3 levels are higher in polyarthritis than in oligoartiritis ?

Response: Thank you very much for your questions and please accept the following explanation common to both questions

The clinical manifestations of the polyarthritis form of JIA are more aggressive, which contributes to the quick diagnosis of the disease. In contrast, oligoarthiritis JIA often develops slowly, which contributes to the chronicity of the inflammatory process. Chronic inflammation seems to be the main reason for the lower levels of both IGF-1 and IGFBP-3 observed in patients with oligoarthritis. It is well  documented that chronic inflammation impairs growth and promotes weight loss in patients [1]. There is a relationship between anabolic process disorders observed in JIA patients and low blood levels of IGF-1, as well as the major regulator of IGF-1 bioavailability, namely IGFBP-3. The results obtained in this study confirm the above relationship. It is believed that the dysregulation of IGF-1-IGFBP-3 axis in the course of JIA originates from the interaction with elevated concentrations of TNF-α, IL-1 and IL-6 [2,1], both at the central and peripheral levels. Specifically, the pro-inflammatory cytokine environment within the pituitary gland contributes to the inhibition of GH secretion, which causes a decrease in the activity of GH receptors in the liver, with subsequent impairment of IGF-1 secretion.

These cytokines reduce the expression of IGF-1 mRNA and interfere with the processes of tyrosine phosphorylation of intracellularly located adapter proteins IRS-1 (insulin receptor substrate 1). They directly block IGF-1 signaling and promote peripheral resistance to IGF-1. Within chondrocytes, TNF-α, IL-6 and IL-1-β may attenuate IGF-1-induced activation of the MAP/ERK 1/2 and phosphatidylinositol-3 kinase pathways, thereby interfering with the function of the epiphyseal plate in children [3-5]. Chronic IL-6 hyperactivity, leads to a reduction in the amount of IGFBP-3 in the blood by inhibiting its biosynthesis in Kupffer cells and increasing its plasma depolymerization [2].

The above mechanisms are briefly described in the Discussion section [p. 13, lines 549-558].

  1. d'Angelo, D.M,; Di Donato, G.; Breda, L.; Chiarelli, F. Growth and puberty in children with juvenile idiopathic arthritis. Pediatr Rheumatol Online J. 2021, 19, 28. doi: 10.1186/s12969-021-00521-5.
  2. De Benedetti, F.; Meazza, C.; Oliveri, M.; Pignatti, P.; Vivarelli, M.; Alonzi, T.; Fattori, E.; Garrone, S.; Barreca, A.; Martini, A. Effect of IL-6 on IGF binding protein-3: a study in IL-6 transgenic mice and in patients with systemic juvenile idiopathic arthritis. Endocrinology 2001, 142, 4818-4826. doi: 10.1210/endo.142.11.8511.
  3. Wong, S.C.; MacRae, V.E.; Gracie, J.A.; McInnes, I.B.; Galea, P.; Gardner-Medwin, J.; Ahmed, S.F. Inflammatory cytokines in juvenile idiopathic arthritis: effects on physical growth and the insulin-like-growth factor axis. Growth Horm IGF Res 2008, 18, 369-378. doi: 10.1016/j.ghir.2008.01.006.
  4. Cirillo, F.; Lazzeroni, P.; Sartori, C.; Street, M.E. Inflammatory diseases and growth: effects on the GH-IGF axis and on growth plate. Int J Mol Sci 2017, 18, 1878. doi: 10.3390/ijms18091878.
  5. Choukair, D.; Hügel, U.; Sander, A.; Uhlmann, L.; Tönshoff, B. Inhibition of IGF-I-related intracellular signaling pathways by proinflammatory cytokines in growth plate chondrocytes. Pediatr Res 2014, 76, 245-251. doi: 10.1038/pr.2014.84.

Reviewer 3 Report

Comments and Suggestions for Authors

The abstract must be revised to enhance the study's comprehensiveness. Classify your design (case-control), and provide additional information about your studied population and numerical data regarding the findings. In the current structure, the abstract is neither informative nor instructive. What was the treatment applied? Additionally, the keywords are too generic and should be replaced with more relevant terms not applied in the title or abstract.

In the methodological section, include the tests applied to assess general blood parameters (Table 1);

It seems that the effectiveness of each treatment was similar, independent of the treatment applied. It would be interesting to include in the table the number of boys/girls who used each medication.

How were the patients stratified by disease forms (e.g., polyarticular JIA)? Were there any challenges in correlating GAAG levels with specific subtypes of JIA?

While GAAGs showed high diagnostic power based on ROC curve analysis, were any limitations or confounding factors observed when using GAAG levels as a diagnostic marker for JIA?

How do GAAG levels compare in diagnostic utility with current standard diagnostic tools like radiographic imaging and clinical evaluation? Could GAAG testing be integrated into current diagnostic protocols?

In tables 2, 3, and 4, the sample size is the same as oligoarthritis and polyarthritis. Explain it.

The sample size of male patients is inadequate, which may hinder strong conclusions.

The authors should include a figure to better demonstrate the metabolic pathways and indicate the potential contributions of their findings.

The conclusion section is too long and must be reduced. Please focus on the main findings.

The authors are invited to include a brief paragraph approaching the main limitations of their study to enable proper data interpretation. Lastly, potential perspectives are also welcome.

Based on your findings, do you believe monitoring GAAG levels could influence treatment decisions, such as the intensity or duration of therapy for JIA patients?

Author Response

Answer to the Reviewer's comment:

We would like to thank the Reviewer for the  evaluation of our article. We are grateful for the important comments, which we fully addressed in the revised manuscript. The text of our manuscript (in each individual part, changes made in blue) has been modified, so as to facilitate its understanding and make it acceptable for publication.

Detailed modifications are presented below:

  1. Comments:

The abstract must be revised to enhance the study's comprehensiveness. Classify your design (case-control), and provide additional information about your studied population and numerical data regarding the findings. In the current structure, the abstract is neither informative nor instructive. What was the treatment applied? Additionally, the keywords are too generic and should be replaced with more relevant terms not applied in the title or abstract.

Response: Following the Reviewer’s recommendation, we have modified the Abstract section [p.1, lines 14-32] and keywords [p. 1, lines 34-35].

  1. Comments:

In the methodological section, include the tests applied to assess general blood parameters (Table 1);

Response: Following the Reviewer’s recommendation, we have modified the Materials and Methods section [p.3, lines 121-126]  and Table 1. [p.4, lines 238].

  1. Comments:

It seems that the effectiveness of each treatment was similar, independent of the treatment applied. It would be interesting to include in the table the number of boys/girls who used each medication.

Response:  Thank you very much for your valuable comment, and at the same time kindly explain. In order to standardise the group of patients, we included only children treated with a combination of oral glucocorticoids, sulfasalazine, and methotrexate. As suggested by the Reviewer, we have modified the description of the treatment of JIA patients in the Materials and Methods section [p.3, lines 131-142].  

  1. Comments:

How were the patients stratified by disease forms (e.g., polyarticular JIA)? Were there any challenges in correlating GAAG levels with specific subtypes of JIA?

Response: Following the Reviewer’s recommendation, we have modified the Materials and Methods section [p.3, lines 111-114] as well as Results section [p.6, lines 248; p.7, lines 272-275].  

  1. Comments:

While GAAGs showed high diagnostic power based on ROC curve analysis, were any limitations or confounding factors observed when using GAAG levels as a diagnostic marker for JIA?

Response: Given the Reviewer’s comments, in the revised manuscript we have added the text fragment regarding the diagnostic usefulness of the circulating GAAG (SC/DS) level in JIA patients [p.13, lines 511-519].

  1. Comments:

How do GAAG levels compare in diagnostic utility with current standard diagnostic tools like radiographic imaging and clinical evaluation? Could GAAG testing be integrated into current diagnostic protocols?

Response: Thank you for your insightful question. Our findings suggest that CS/DS blood levels offers a distinct advantage as a potential biomarker for JIA diagnosis. Unlike radiographic imaging which primarily identifies cartilage destruction at later stages CS/DS levels reflect systemic metabolic changes in cartilage extracellular matrix turnover, providing a biochemical perspective an disease activity. Moreover, alterations in CS/DS levels appear to occur early in disease process, suggesting that CS/DS testing could facilitate earlier diagnosis compared to conventional imaging methods. However, its full adoption will require further validation and standardization of measurement techniques.

In response to a reviewer's question, we have modified the Discussion section [p.13, lines 520-537].

  1. Comments:

In tables 2, 3, and 4, the sample size is the same as oligoarthritis and polyarthritis. Explain it.

Response: Thank you very much for your comment, and at the same time kindly explain. In Tables 2, 3 and 4 (first column of results – control subjects), the sample size is the same for oligoarthritis and polyarthritis because it refers to the total content of GAAGs (CS/DS), IGF-1 and IGFBP-3 in the blood of control subjects. In order to improve readability, the tables have been modified. These values are presented as a combined reference group for comparison with the respective JIA subtypes [p.6, lines 248; p.7, lines 288;  p.8, lines 325].

  1. Comments:

The sample size of male patients is inadequate, which may hinder strong conclusions.

Response: We fully agree with the reviewer's comment. Drawing strong conclusions requires a larger sample size, particularly a more balanced representation of male patients. The relatively small   number of male patients in this study reflects the low prevalence of JIA among children in Poland (5–6.5 per 100,000 children) and the higher incidence of JIA in females.  We have added this information to the Conclusions section [p.14, lines 589-593].

  1. Comments:

The authors should include a figure to better demonstrate the metabolic pathways and indicate the potential contributions of their findings.

Response: We would like to thank you very much for pointing the possibility of graphical representation of metabolic pathways in which our research fits in. These studies are part of a project investigating metabolic changes in the extracellular matrix components in children with JIA. The figure proposed by the Reviewer will enrich the review paper on this topic, that our team is currently preparing.

  1. Comments:

The conclusion section is too long and must be reduced. Please focus on the main findings.

Response: Following the Reviewer’s recommendation, we have modified the Conclusion section [p.14, lines 582-593].

  1. Comments:

The authors are invited to include a brief paragraph approaching the main limitations of their study to enable proper data interpretation. Lastly, potential perspectives are also welcome.

Response: In accordance with the reviewer's comment, we have added information about the limitations of the research and potential perspectives to the work.

  1. Comments:

Based on your findings, do you believe monitoring GAAG levels could influence treatment decisions, such as the intensity or duration of therapy for JIA patients?

Response: Thank you very much for your question. Our data suggest that circulating CS/DS levels could be useful for monitoring the clinical status of treated JIA patients. However, it should be emphasized that our study concerning the utility of CS/DS blood concentration both JIA diagnosis and monitoring of the efficacy of the disease treatment is preliminary. Additional examinations in larger patient groups are necessary to better assess the diagnostic and therapeutic implications of the parameter, especially in relation to radiological imaging results and clinical manifestation of the disease. Another factor determining the utility of the circulating CS/DS is the method of measurement. In our study, we used a more complex procedure, including the isolation of these compounds and their quantification by hexuronic acid assay. This procedure can be greatly simplified to a direct reaction of chondroitinase ABC on CS/DS in plasma (serum) samples, with measurement of absorbance at 232 nm, which corresponds to  the maximum light absorbance for unsaturated disaccharides, being products of specific enzymatic degradation of these glycan compounds. However, a full analytical characteristic of this simplified method remains to be performed. On the other hand, further research should focus on identifying a structural signature of circulating CS/DS, especially related to the sulfation pattern of these compounds, which may be unique for JIA. Such a feature could facilitate the development of an immunoassay, making the analyte more practical for use  in laboratory practice. We have included this explanation in the Discussion section [p.13, lines 511-537].

Round 2

Reviewer 3 Report

Comments and Suggestions for Authors

The authors have sufficiently addressed my comments and concerns. I would like to express my gratitude for your thoughtful consideration and respectful feedback. I am in favor of publishing the manuscript in its current form.